# Dental Management of Maxillofacial Ballistic Trauma

**DOI:** 10.3390/jpm12060934

**Published:** 2022-06-05

**Authors:** Edoardo Brauner, Federico Laudoni, Giulia Amelina, Marco Cantore, Matteo Armida, Andrea Bellizzi, Nicola Pranno, Francesca De Angelis, Valentino Valentini, Stefano Di Carlo

**Affiliations:** 1Department of Oral and Maxillofacial Sciences, Sapienza University of Rome, Via Caserta 6, 00161 Rome, Italy; edoardo.brauner@uniroma1.it (E.B.); federico.laudoni@gmail.com (F.L.); marcocantore2@gmail.com (M.C.); matteoarmida@virgilio.it (M.A.); and.bellizzi@gmail.com (A.B.); nicola.pranno@uniroma1.it (N.P.); francesca.deangelis@uniroma1.it (F.D.A.); valentino.valentini@uniroma1.it (V.V.); stefano.dicarlo@uniroma1.it (S.D.C.); 2Implanto-Prosthetic Unit, Policlinico Umberto I, Viale Regina Elena 287b, 00161 Rome, Italy; 3Oncological and Reconstructive Maxillo-Facial Surgery Unit, Policlinico Umberto I, Viale del Policlinico 155, 00167 Rome, Italy

**Keywords:** maxillofacial ballistic injuries, gunshot injuries, low-velocity ballistic wounds, high-velocity ballistic wounds, reconstructive surgery, dental rehabilitation, fixed implant-supported prosthesis

## Abstract

Maxillofacial ballistic trauma represents a devastating functional and aesthetic trauma. The extensive damage to soft and hard tissue is unpredictable, and because of the diversity and the complexity of these traumas, a systematic algorithm is essential. This study attempts to define the best management of maxillofacial ballistic injuries and to describe a standardized, surgical and prosthetic rehabilitation protocol from the first emergency stage up until the complete aesthetic and functional rehabilitation. In low-velocity ballistic injuries (bullet speed <600 m/s), the wound is usually less severe and not-fatal, and the management should be based on early and definitive surgery associated with reconstruction, followed by oral rehabilitation. High-velocity ballistic injuries (bullet speed >600 m/s) are associated with an extensive hard and soft tissue disruption, and the management should be based on a three-stage reconstructive algorithm: debridement and fixation, reconstruction, and final revision. Rehabilitating a patient with ballistic trauma is a multi-step challenging treatment procedure that requires a long time and a multidisciplinary team to ensure successful results. The prosthodontic treatment outcome is one of the most important parameters by which a patient measures the restoration of aesthetic, functional, and psychological deficits. This study is a retrospective review: twenty-two patients diagnosed with outcomes of ballistic traumas were identified from the department database, and eleven patients met the inclusion criteria and were enrolled.

## 1. Introduction

The maxillofacial traumatology presents a great etiological, clinical, and therapeutic diversity. A ballistic injury, caused by a firearm or by the sudden explosion of ammunition, is rarely neat, clean, and predictable: due to a significant soft tissue and bone loss from the impact, this type of facial trauma represents a challenge for reconstructive surgery and for the final aesthetic and functional rehabilitation [1]. The damage that a bullet creates is unpredictable, and the management in planning and reconstructing a traumatic defect requires a multi-step approach and a multidisciplinary team [2,3,4]. In addition to the extensive damages to both soft and hard tissues, gunshot wounds cause deleterious physical and psychiatric effects, representing a complex set of challenges for the maxillofacial surgeon, the oral surgeon, the prosthodontist, and, sometimes, for the psychologist or psychiatrist [5,6,7]. In most cases, maxillofacial gunshot injuries may not result in a life-threatening trauma but are frequently associated with severe aesthetic modifications, masticatory dysfunction, or speech- or deglutition-related issues: rehabilitating such patients can be frustrating and challenging, so we need to generalize and standardize our treatment plan [3]. The rehabilitation of patients who attempted suicide due to psychological trauma and to the devastating outcome of self-inflicted ballistic trauma must consider both psychologic and anatomical problems: positive emotional support is as important as the surgical and prosthodontic management of the defects [6,8,9]. There is a scarcity of reports related to the management of gunshot injuries in the maxillofacial region [10]. However, wartime-related injuries are mostly described in the literature, making it difficult to apply these information to civilians due to dissimilarities in wound contamination and wounding potential of firearms and ammunition: gunshot injuries in civilians should be considered as distinct entities [11,12]. The nature and the severity of a ballistic trauma depend on many factors: the caliber of the weapon; the shooting distance; and the size, the shape, the velocity of the shrapnel and its fragments but also the characteristics of the tissue through elasticity, density, and their anatomical relationships [5,13,14]. Because of the diversity and the complexity of facial gunshot injuries, a systematic algorithm is essential [15]; it is therefore also essential, for the proper treatment of these patients, not only the knowledge of general traumatology but also of the technical aspects concerning firearms, the composition and the shape of the cartridges, ballistics, and characteristics of gunshot wounds [16]. The aim of this study is to define the best management of maxillofacial ballistic injuries and to describe a standardized, surgical, and prosthetic, rehabilitation protocol from the first emergency stage up until the complete aesthetic and functional rehabilitation.

### 1.1. Epidemiology

Ballistic traumas are not so uncommon. Firearm-related injuries represent 16% of the general traumatology and are one of the main causes of death and disability [16]. Recent studies state that approximately 32,000 deaths and 67,000 injuries are caused by firearms each year [1]. Male patients are most frequently involved, representing 80% of the gunshot wound patient’s population. Approximately two-thirds of gunshot injuries are caused by a single gunshot wound, resulting in 11% mortality within 24 h after trauma [15]. Ballistic injuries are responsible for 2–6% of facial fractures [17]; indeed, head and neck are one of the most commonly affected areas: more than 50% of attempted suicides, 14% of assaults, and 12% of accidental injuries [2,10]. A retrospective review of nearly 4100 gunshot wounds observed that about 6% involved the face [15]. The most common mode is ballistic projectiles and gunshot wounds (41.4%), followed by splinter and shrapnel injuries from improvised explosive devices blasts (39.2%) [18].

### 1.2. General Characteristics of Maxillofacial Gunshot Injury

Generally, weapon injuries are categorized as low-velocity and high-velocity injuries depending on the speed of impact. Low-velocity wounds, with a projectile velocity of less than 600 m/s (<2000 fts), are classically caused by handguns and shotguns and are therefore more common in the civilian population; this injury is usually less severe and non-fatal. High-velocity wounds, caused by military or hunting weapons, with a projectile velocity superior to 600 m/s (>2000 fts), are associated with extensive hard and soft tissue disruption and are characterized by a surrounding area of necrotic tissue caused by a compromised blood supply and sepsis: the outcome is usually fatal [2,12,19,20].

Rose et al., in a retrospective review, reported that with high-energy weapons, all fractures were complete and comminuted; by comparison, for low-energy weapons, 60% of the fractures were incomplete, and only 22% were multi-fragmentary [21]. Clark et al. distinguished between avulsive and non-avulsive injuries based on the energy transferred by the bullet. Avulsive injuries provoked by high-intensity ballistic traumas cause a significant destruction and loss of both hard and soft tissue, resulting in a greater reconstructive challenge. In non-avulsive injuries, as a result of low-intensity ballistic trauma, the majority of soft tissue remains [22]. A low-velocity projectile that passes through human tissue can necrotize both the entry and exit margin and a little bit of circumferential tissue. A high-velocity projectile, indeed, may cause cavitation in crossed tissue, and it can create greater destruction and more extensive necrosis [12]. We can differentiate two areas of projectile-tissue interaction: the permanent and the temporary cavity. According to Harvey, Janzon, Sellier, and Kneubuehl, the temporary cavity is the most important factor in high-velocity ballistics wound because the damage is far less predictable. In general, we can state that the capability of a projectile is directly related to the amount of kinetic energy dissipated in the tissue, where M and V represent, respectively, the mass and the velocity of the projectile. As explained by the formula, a doubling of velocity can quadruple the energy of the projectile [11,12,23,24]. The classification of maxillofacial firearm injuries also depend on the anatomical structures that are involved: Bènateau et al.’s classification distinguishes the central from the lateral facial areas, dividing each one into lower (mandible), middle (maxilla), and upper units; elasticity, density, and anatomic relationships of the involved tissue strongly affect wound character [15,20]. Brauner et al. introduced a new classification to define dentofacial traumas based on seven parameters: number of teeth lost (T1 < 2, T2 2–3, T3 4–5, T4 > 5), upper/lower maxilla (U/L), alveolar/basal bone (Alv/B), gingival tissues (G), soft tissues (S), adult/child (a/c), and reconstructed patient (R). This classification is useful for precisely define the entity of the trauma and to establish the best therapy choice, making the prognosis of these patients more predictable [25].

## 2. Materials and Methods

This study is a single-institution, retrospective review. Inclusion criteria are: (1) patients with outcome of ballistic trauma in maxillofacial region; (2) patients prosthetically rehabilitated in Implantoprosthesis Unit of Head and Neck Department in “Sapienza” University of Rome, Policlinico “Umberto I”; and (3) complete clinical and radiological documentation. All patients gave their informed consent for inclusion. The study was approved by the Ethics Committee of “Sapienza” University of Rome (N. 111/2022 Prot. N. 0000867, 18 May 2022). Twenty-two patients diagnosed with outcomes of ballistic traumas were identified from the department database. Two of these patients who attempted suicide died because of a second self-inflicted injury. Three of these patients were excluded because they were the result of a self-inflicted shotgun injury, and due to psychological trauma and insufficient compliance, they did not complete dental rehabilitation. Self-inflicted submental gunshot wounds are notoriously difficult to treat and prone to complication. In three cases, the contamination of the surgical site and the consequent recurrent infections did not allow a correct management and prosthetic rehabilitation of the patient. Two patients were excluded because the documentation was insufficient. One patient underwent a devastating high-velocity ballistic trauma, so the damages and the contamination of the wound was massive and required a second surgical procedure before final dental rehabilitation. Eleven patients met the inclusion criteria and were enrolled in this study. The age and gender of patients, injury patterns based on bullet-speed, the anatomic sites involved based on Brauner’s classification, and the type of resection, reconstruction, and dental rehabilitation are shown in Table 1.

## 3. Results

### 3.1. Sample Characteristic

Eleven patients met the inclusion criteria and were enrolled. Most patients were male (90.9% of the sample): the mean age was 30 years. Seven patients reported an injury caused by a high-velocity ballistic trauma (63.6% of the sample), while the remaining five were involved in a low-velocity ballistic trauma (45.4% of the sample). The main cause of the injuries was a military weapon (six cases), and three patients sustained a shotgun wound, while in two cases, the responsible weapon was a handgun. The lower maxilla was involved in most cases (63.6% of the sample), the maxilla was involved in three cases (27.2% of the sample), and in only one patient was it necessary to rehabilitate both jaws. In all eleven patients, almost five dental elements were lost (T3/T4 according to Brauner et al., classification [25]).

### 3.2. Surgical Treatment Details

All eleven patients, who arrived with an injury resulting from a ballistic trauma, were initially managed in according with ATLS protocol (Advanced Trauma Life Support): identification of life-threatening injuries, stabilization of the patient and airway control, bleeding control, and maintenance of adequate circulation (ABC) [12,15,16,22,26]. Then, all patients underwent the first surgical stage, ideally performed 24–48 h after the injury, consisting of debridement of foreign material, nonviable tissue, teeth, and bone; maxillomandibular fixation; and ORIF, soft tissue closure, and CT confirmation to plan the definitive hard and soft tissue reconstruction and to plan dental and facial prosthetic rehabilitation in conjunction with flap selection [22,26,27]. Only five patients (45.4% of the sample) underwent the definitive reconstruction (Stage 2): in three patients, it was necessary to perform a microsurgical reconstruction, two patients with a free fibula flap, and one patient with a free iliac crest flap, while in two cases, a GBR was sufficient. The second stage was ideally performed within 2 weeks after the injury [22]. The use of bone grafts, especially the free fibula graft, offers important advantages for implant positioning and optimal prosthetic rehabilitation in absence of adequate native bone [28,29,30].

### 3.3. Dental Rehabilitation Details

The last stage is the final revision, ideally performed >3 months after the reconstruction [22]: it includes flap debulking, scar revision, dental rehabilitation, final oral commissuroplasty, and adjunctive cosmetic measures. One patient with high-velocity ballistic trauma is still being evaluated: the bone defect and the massive wound contamination are too extensive, so we are planning a new reconstructive procedure before dental rehabilitation. Eleven patients underwent dental rehabilitation. At first, all patients were rehabilitated with partially or totally removable prosthesis to recover the functionality (occlusion, chewing, joint function, swallowing, and phonation) and aesthetics (facial asymmetries, soft tissue defects, lost teeth) of the stomatognathic system and to plan the design of the future fixed prosthesis. Only six patients decided to carry on the treatment with the final implant-retained prosthesis. Based on the previous temporary partially removable prosthesis, a custom-made radiographic guide was manufactured, and the CBCT was performed to plan the correct implant design and positioning. Then, the custom-made guide was used during the surgery to allow the implants positioning. The implant surgery was performed under local anesthesia. A total amount of 36 implants were inserted (manufactured by: Zimmer Dental Inc. 1900 Aston Avenue Carlsbad, CA 92008 USA). For all the cases, it was decided to utilize a titanium bar that connected the implants to the metal substructure of the prosthesis, ensuring a better distribution of the masticatory forces. The final implant-retained prosthesis consisted of three different components: a titanium bar screwed on implants, a primary metal structure assembled on the base, and a secondary structure that reproduced teeth with or without gingiva.

Three explanatory cases of implant rehabilitation are shown in Figure 1, Figure 2, Figure 3, Figure 4, Figure 5, Figure 6, Figure 7, Figure 8 and Figure 9 to demonstrate the same prosthetic protocol despite the different localization and type of residual defect after maxillofacial surgery.

In the first case, a low-velocity ballistic trauma caused the loss of teeth 4.3, 4.2, 4.1, 3.1, and 3.2 and the loss of a large portion of gingiva and basal bone in mandibular area; an anterior alveolar resection of the mandible was performed, and the definitive reconstruction consisted of a surgical bone graft taken from the mandibular angle. Nine months after surgery bone reconstruction, the implant surgery was performed, and four implants, namely “Zimmer Trabecular Metal”, were inserted in anterior lower maxilla. In the same area, a fornix depth was performed. After four months, healing screws were inserted, and three months later, the patient finalized the treatment with five metal/ceramic crowns (primary metal structure and secondary ceramic structure) (Figure 1, Figure 2 and Figure 3).

**Figure 1 jpm-12-00934-f001:**
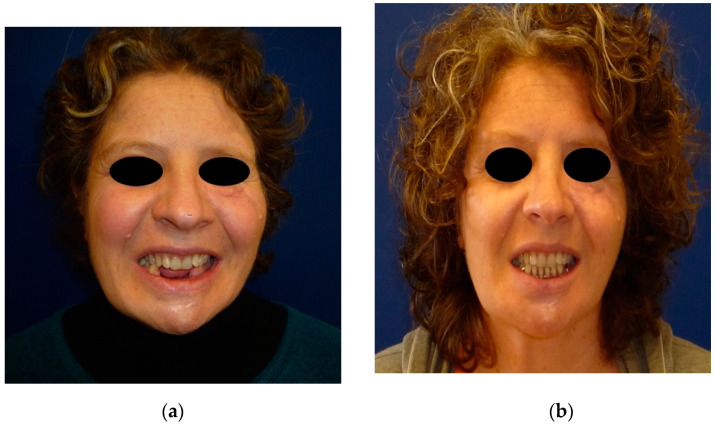
A 41-year-old female patient wounded by a low-velocity injury weapon. (**a**) Before prosthetic treatment; (**b**) after prosthetic treatment.

**Figure 2 jpm-12-00934-f002:**
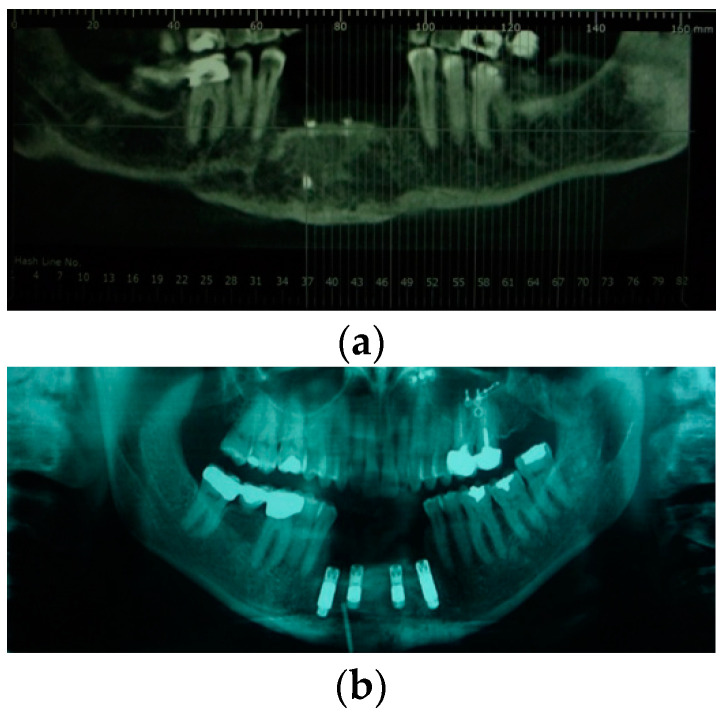
(**a**) Radiography imaging before prosthetic rehabilitation; (**b**) Radiography imaging after implant surgery.

**Figure 3 jpm-12-00934-f003:**
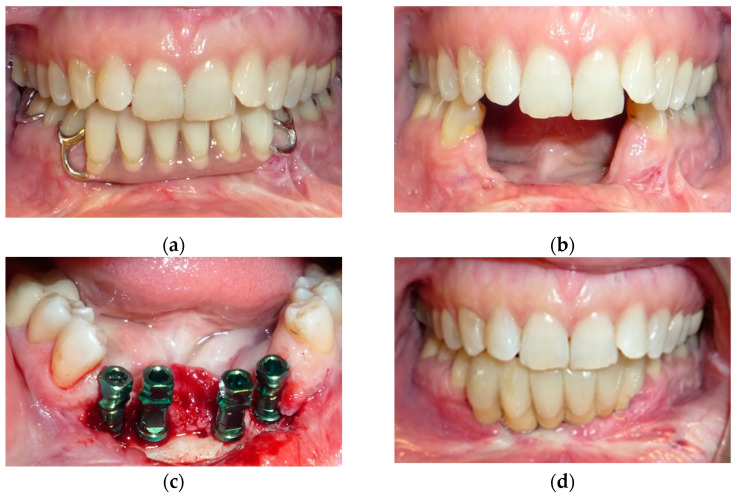
Intraoral view: (**a**) temporary resin removable prosthesis; (**b**) after a fornix depth surgery; (**c)** implant surgery; (**d**) final implant-retained prosthesis.

In the second case, a high-velocity ballistic trauma caused the destruction of right premaxilla and the loss of dental elements 1.1, 1.2, 1.3, 1.4, 1.5, and 2.1 and adjacent soft tissue. The patient presented an extensive hard and soft tissue disruption; he lost a large portion of labial soft tissue, and he showed a retracting and hypertrophic scar in the traumatized zone. The first surgery was performed: it consisted of a right emimaxillectomy followed by the second reconstructive surgery with osteomyocutaneous free fibula flap. Four months after definitive surgery and reconstruction, the implant surgery was performed, and six implants, namely “Zimmer Trabecular Metal”, were inserted in right emimaxilla. In a second step, the reconstruction of the upper lip using an Abbé mucocutaneous flap was performed. After four months, we inserted healing screws, and six months later, the patient finalized treatment (Figure 4, Figure 5 and Figure 6).

**Figure 4 jpm-12-00934-f004:**
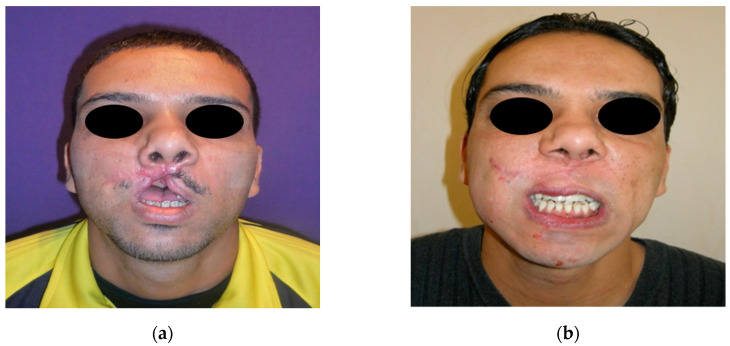
A 29-year-old male patient wounded by a high-velocity injury weapon. (**a**) Before prosthetic treatment; (**b**) after prosthetic treatment.

**Figure 5 jpm-12-00934-f005:**
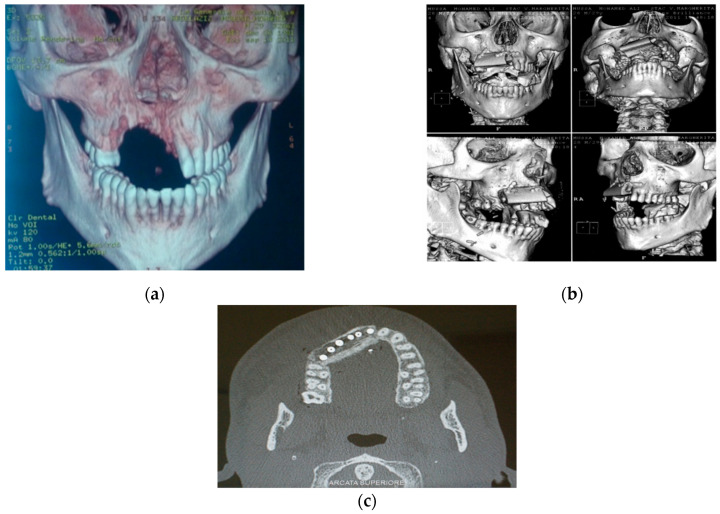
(**a**)CT scan before reconstruction surgery and prosthetic rehabilitation; (**b**) 3D CT scan after reconstruction surgery with osteomyocutaneous free fibula flap; (**c**) CT scan after placement of six implants.

**Figure 6 jpm-12-00934-f006:**
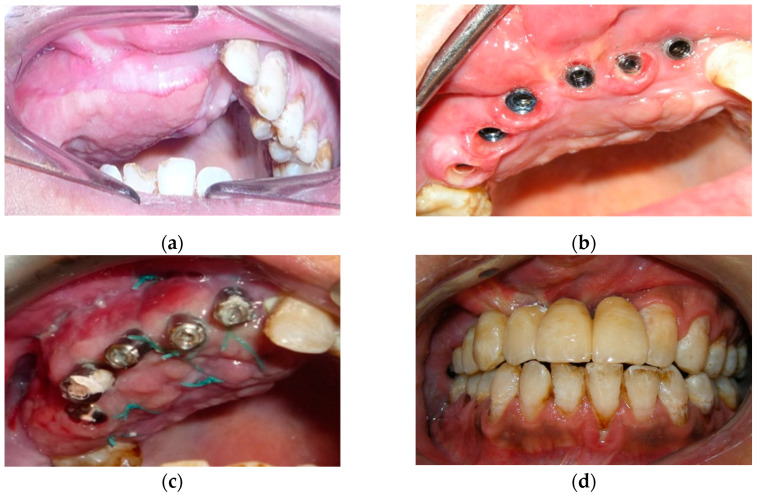
Intraoral view: (**a**) before treatment; (**b**) dental implant placed; (**c**) dental implant placed after a fornix depth; (**d**) final implant supported prosthetic rehabilitation.

In the last case, a high-velocity ballistic trauma caused the loss of alveolar bone in different areas of the jaw with loss of teeth in the first, second, and fourth quadrant and of adjacent soft tissue. The patient presented an extensive soft tissue depression of the cheek with a retracting and hypertrophic scar in the trauma zone. During the surgery, the traumatized alveolar bone zone was cleaned; then, an alveolar resection of quad. I and quad. IV was performed followed by the second reconstructive surgery of the area, carried out with osteomyocutaneous free iliac crest flap. Four months after first surgery, ten implants, namely “Zimmer Trabecular Metal”, were inserted in sites to be rehabilitated. After four months, healing screws were inserted, and six months later, the patient finalized the implant-supported prosthetic rehabilitation. In both the second and third case, the final prosthesis consisted of a titanium primary structure and a composite-coated secondary structure that reproduced teeth with or without gingiva (Figure 7, Figure 8 and Figure 9).

**Figure 7 jpm-12-00934-f007:**
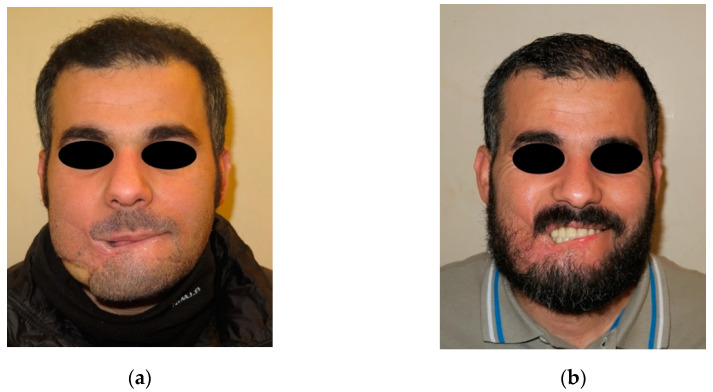
A 31-year-old male patient wounded by a high-velocity injury weapon. (**a**) Before prosthetic treatment; (**b**) after prosthetic treatment.

**Figure 8 jpm-12-00934-f008:**
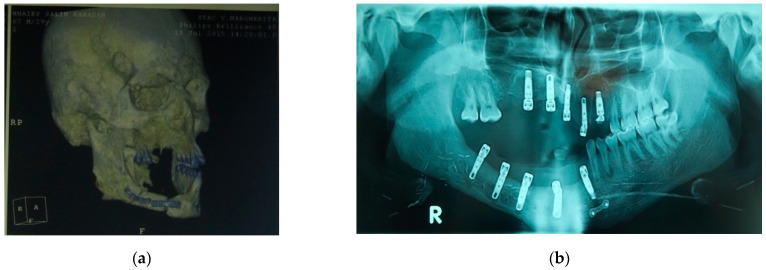
(**a**) 3D imaging before prosthetic treatment; (**b**) Rx imaging after implant surgery. Three-dimensional, (3D).

**Figure 9 jpm-12-00934-f009:**
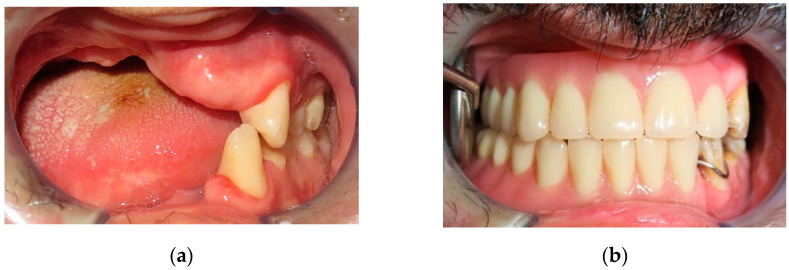
Intraoral view: (**a**) before prosthetic treatment; (**b**) temporary resin removable prosthesis; (**c**) a titanium base screwed on implants; (**d**) final implant supported prosthetic rehabilitation.

## 4. Discussion

Maxillofacial ballistic trauma is associated with complex impairment evaluation due to its biological and psycho-social consequence. Individuals’ psychological and social well-being must also be considered. According to Figueiredo et al., these complex and devastating traumas cause post-traumatic stress disorders, associated with behavioral or emotional health (depression and anxiety) and other risk (interpersonal relationship functioning): rehabilitating such patients is a priority [31]. Because of the diversity and the complexity of these traumas, a systematic algorithm is essential [15].

### 4.1. Surgical Treatment

Patients with a maxillofacial gunshot wound must be initially managed in accordance with the ATLS protocol [12,15,16,22,26]. The timing of definitive reconstruction remains an area of continuing debate. There are two schools of thought in literature on principles of management of maxillofacial ballistic injuries. The older, three-phased approach (Futran et al.) consists of immediate debridement, fracture stabilization, and primary wound closure, thereafter followed by delayed hard and soft tissue reconstruction and subsequently by revision of residual deformities of the oral cavity: for this last phase, flap debulking, dental implant placement, and scar revisions are important components. The newer trend consists in an early and aggressive approach with open surgical reduction and fixation and reconstructive procedures carried out in one step with the soft tissue’s debridement and closure [5,7,15,19,21,22,26,32,33,34,35]. The advantages of an early and aggressive reconstruction are an increased tissue mobilization, a reduced tissue fibrosis, and a reduced period of facial deformity. Due to the severity and to the uncertain margins of the injuries, we preferred the conservative approach and primary non-operative management; the reconstructions are often performed as secondary intentions. According to Kaufman et al. [15], we claim that the prolonged period decreases infection rates, reduces necrotic debris, and allows the surgeon to obtain a better understanding of the extent of the injury to formulate a definitive treatment plan, but in the delayed approach, there is an increased incidence of wound contracture, which results in a more severe structural and functional deformity. Oren et al. [19] unintentionally demonstrated that a delayed treatment of high-velocity injuries may have contributed to a critical revascularization period, resulting in improved healing and a decrease of postoperative morbidity and complications. However, Vasconez et al. observed similar infection rates between delayed and immediate gunshot wound reconstruction [15]. Risk factors for infection included a delayed wound management, lack of an adequate wound management, a wound size between 1–2 cm, and incorrect wound care [12]. An adequate wound debridement and a correct stabilization of the fracture site are the mainstays of treatment: the fracture site should be considered highly contaminated, and while many fragments will appear viable, the wounds must be washed out, and if necessary, fragments should be excised to minimize the risk of infection [33,34,35]. Surgical reconstruction of discontinue defects often involves the use of grafts dictated by the size of the defect and surgeon’s preference. Bone grafting is reserved for small defects with an adequate soft-tissue vascularization; larger defects and smaller defects with insufficient lining require composite free tissue transfer. There is not a single treatment that could be addressed for reconstruction and the evaluation of the patient’s situation is essential for the treatment planning [10,21,26,36]. However, the immediate aggressive reconstruction or delayed secondary reconstruction depends on the patient’s condition, the wound status and the surgeon’s judgment.

### 4.2. Dental Rehabilitation

The prosthodontic treatment outcome is one of the most important parameters by which a patient measures the restoration of aesthetic, functional, and psychological deficits caused by trauma [10,31,37,38,39]. In our opinion, the best treatment is an implant-retained prosthesis: the literature agrees that a fixed prosthesis gives patients greater satisfaction than a mobile one. Implant rehabilitation allows a proper retention of removable prosthesis with the decrease in the load on soft tissue. Commonly, in patients with weapon injuries, the severe ridge irregularities, the inadequate soft tissue support, and the high risk of infection can adversely affect the survival, the retention, and the stability of an implant-supported prosthetic. In our opinion and according to Awadalkreem et al. [3], a fixed hybrid prostheses is the best prosthetic treatment option for our patient owing to the easier handling, quick treatment duration, metal framework that ensures better force distribution, and the presence of hygienic space under the prosthesis flanges that ensures the washing action of saliva and prevents food and plaque accumulation. Further, the hybrid design compensates the severe soft and hard tissue loss and provides sufficient lip support: this advantage eliminates the need for bone grafting and its risk factors [3]. Implant-retained prosthesis with acrylic resin teeth and metal substructures have been described: the metal substructure helps to enhance the strength of the acrylic resin, especially when there is a distal cantilever extension in the prosthesis [10]. In general, several factors should be taken into consideration on the implant survival: surgeon’s experience, bone quality, bone topography, and the technical aspects such as implant length, diameter, and primary stability [3,33]. The correct osseointegration and the survival of dental implants can heavily be influenced by side effects such as xerostomia, persistent hyposalivation, fibrosis of soft tissue, delayed healing, reduced angiogenesis, and, in our opinion, the most important factor in ballistic trauma, the wound contamination. The ballistic wounds become contaminated during the projectile penetration and secondarily through their exposed surface [20]. As widely stated above, the major spread of contamination is along tissue planes, and adequate wound debridement is essential in the management of ballistic wounds: knowledge of the extent of contamination is necessary, and we always must consider the risk of secondary infection. In our experience, the fixed hybrid prosthesis has guaranteed the best follow-up because of the easier handling, quick treatment duration, a better force distribution ensured by the metal structure, and the presence of the space under the prosthesis flanges that facilitates the maintenance of hygiene by preventing food and plaque accumulation. Further, the hybrid design compensates for the severe soft and hard tissue loss, key features of such trauma, and provides sufficient lip support: this advantage includes eliminating the need for bone grafting and its risk factor.

## 5. Conclusions

Maxillofacial ballistic injuries require a multidisciplinary and multistep approach to achieve the best rehabilitation (Figure 1). The knowledge of the general traumatology and of ballistics is therefore essential. In addition, a standardized and systematic protocol allows to treat this kind of patients in the same way, guaranteeing an effective result despite their complexity and diversity. All maxillofacial ballistic injuries must be initially managed in accordance with the algorithmic advanced trauma life support (ATLS protocol). In low-velocity injuries, the management can be based on early, definitive, and aggressive surgery and reconstruction; high-velocity maxillofacial injuries represent a significant reconstructive challenge, and although the existing literature suggests early and aggressive intervention, the evidence is limited, and we prefer a three-stage reconstructive algorithm: debridement and fixation, reconstruction, and final revision. The final revision always includes oral rehabilitation: improving the quality of life of these patients is crucial, and adequate dental implant rehabilitation improves esthetic and functional results, increasing the patient’s self-esteem. This study has attempted to define the best management of maxillofacial ballistic injuries and to underline the importance of optimal oral rehabilitation for this kind of patients.

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
