# Peer review of "Dental Management of Maxillofacial Ballistic Trauma"

_jpm, 2022, doi:10.3390/jpm12060934_

Round 1

Reviewer 1 Report

Dear Authors,

The type of traumatic event, the ballistic, concerning the orofacial trauma is an unusual topic in periodic journals. The authors present an exciting issue according to real life!

In this sense, I should perform several suggestions to increase the quality of the paper. 

The material section is a weak section of the paper, although it should be clarified: 1) is the study designing a retrospective study or a series of cases?; 2) Was consent obtained?;  3) the inclusion and exclusion criteria should be point-out; 4) table 1 –  the legend should explain the tables' content, as well as the symbols; 5) the criterium of ballistic velocity should be referenced in the bibliography; 6) the injuries classification should be present, the authors classified the injuries following anatomic topography, and it should be a method to be presented.

The methodology was in line with the ATLS protocol, and the authors should reference

The authors write "to recover the functionality and aesthetics of the stomatognathic system." The authors should consider this approach and discuss it according to the sequels' impairment or personal evaluation.

The discussion section should focus on the impairment concerning the Maxillofacial Ballistic Trauma. I suggest the paper: Figueiredo, C., Coelho, J., Pedrosa, D., Caetano, C., Corte-Real, F., Vieira, D. N. & Corte-Real, A. (2019). Dental evaluation specificity in orofacial damage assessment: a serial case study. Journal of Forensic and Legal Medicine, 68,101861. doi:10.1016/j.jflm.2019.101861. It will improve the discussion section.

The authors should rewrite the conclusion section. "The prosthetic rehabilitation has the greatest impact on the patient's quality of life: an implant-supported prosthesis allows the restoration of aesthetic, functional and psychological aspects." (lines 355-357) was not studied. The authors do not explore the quality of life, so it is not a conclusion of the present study. The conclusion section is not a viewpoint. The sentence "In our opinion, the fixed hybrid prosthesis guarantees the best follow-up because of the easier handling,  quick treatment duration, a better forces distribution ensured by the metal structure and the presence of the space under the prosthesis flanges that facilitates the maintenance of hygiene by preventing food and plaque accumulation." (lines 357-361) is not a conclusion. It is probably a point of view for the discussion section.

Author Response

Response to Reviewer 1 Comments

Point 1: The material section is a weak section of the paper, although it should be clarified: 1) is the study designing a retrospective study or a series of cases?; 2) Was consent obtained?;  3) the inclusion and exclusion criteria should be point-out; 4) table 1 –  the legend should explain the tables' content, as well as the symbols; 5) the criterium of ballistic velocity should be referenced in the bibliography; 6) the injuries classification should be present, the authors classified the injuries following anatomic topography, and it should be a method to be presented.

Response 1:

  • the study designing is a retrospective study, as I clarified in the first part of section “Materials and Methods”: << This study is a single-institution retrospective study performed on patients with outcome of maxillofacial ballistic trauma treated in Implantoprosthesis Unit of Head and Neck Department in “Sapienza” University of Rome, Policlinico “Umberto I”>>.
  • Yes, the consent was obtained by subjects. I add this information in the “Materials and Methods” section.
  • I improved the “Materials and Methods” section, now the Inclusion criteria will be clearer.
  • Ar reguard the Table 1, I specify the contents of the table in Material and Method section. I modified the anatomic sites involved section introducing Brauner’s classification. *I have changed the wording “Mechanism of injury” in “Injury patterns”, I think is more correct.
  • The reference of injury classification has alredy been presented in the section 1.2 “General characteristics of maxillofacial gunshot injury” (References: 2, 12, 19, 20).
  • In 1.2 section we describe differents type of classifications of these traumas discussed on literature, however in our study we classified patient’s trauma based on bullet speed and according to Brauner’s classification (I add it in Table 1).

Point 2: The methodology was in line with the ATLS protocol, and the authors should reference

Response 2: I agree. I’ve added these reference in 3.2 section, line 172.  

Point 3: The authors write "to recover the functionality and aesthetics of the stomatognathic system." The authors should consider this approach and discuss it according to the sequels' impairment or personal evaluation.

Response 3: I modified the sentence "to recover the functionality and aesthetics of the stomatognathic system" with “At first, all patients were rehabilitated with partial or total removable prosthesis to recovers the functionality (occlusion, chewing, joint function, swallowing and phonation) and aesthetics (facial asymmetries, soft tissue defects, lost teeth) of the stomatognathic system and to plan the design of the future fixed prosthesis.”

Point 4: The discussion section should focus on the impairment concerning the Maxillofacial Ballistic Trauma. I suggest the paper: Figueiredo, C., Coelho, J., Pedrosa, D., Caetano, C., Corte-Real, F., Vieira, D. N. & Corte-Real, A. (2019). Dental evaluation specificity in orofacial damage assessment: a serial case study. Journal of Forensic and Legal Medicine, 68,101861. doi:10.1016/j.jflm.2019.101861. It will improve the discussion section.

Response 4: Thank you for the reference. It was useful to me and I have implemented the discussion section. I also add this reference in bibliography. I hope will be better now.

Point 5: The authors should rewrite the conclusion section. "The prosthetic rehabilitation has the greatest impact on the patient's quality of life: an implant-supported prosthesis allows the restoration of aesthetic, functional and psychological aspects." (lines 355-357) was not studied. The authors do not explore the quality of life, so it is not a conclusion of the present study. The conclusion section is not a viewpoint. The sentence "In our opinion, the fixed hybrid prosthesis guarantees the best follow-up because of the easier handling,  quick treatment duration, a better forces distribution ensured by the metal structure and the presence of the space under the prosthesis flanges that facilitates the maintenance of hygiene by preventing food and plaque accumulation." (lines 357-361) is not a conclusion. It is probably a point of view for the discussion section.

Response 5: I modified the conslusion section, adding this paragraph in the discussion section.

Reviewer 2 Report

Well written and nicely illustrated case serie article however does not provide new interesting scientific data 

Author Response

About the references, they were all mentioned in the text and they were all useful to me in writing this manuscript. 

About the English language,  I have resubmitted the article to a native English speaker. 

Reviewer 3 Report

Please have this paper for extensive English proofreading.

The aim in the abstract section is different from the one stated in the introduction.

The formatting of the paper is not in line with the journal template, for example:
Headings should not be included in the abstract (e.g. Background:, Methods:); keywords should not be numbered; references are not properly formatted 

Add City/Country for Implant Manufacturers

This article is too generally written; it is not obvious what are the elements from the case presentations that led to the conclusion and results. In addition, the treatment protocol is overly simplified.

Avoid repetitions throughout the text.

Conclusions must be shortened, simplified and also must avoid repetitions.

Author Response

Response to Reviewer  Comments:

Point 1: Please have this paper for extensive English proofreading.

Response 1: About the English language, I have resubmitted the article to a native English speaker. 

Point 2: The aim in the abstract section is different from the one stated in the introduction.

Response 2: I modified the abstract. In the abstract section I wrote “this study wants to define the best management of maxillofacial ballistic injuries and to describe a standardized, surgical and prosthetic, rehabilitation protocol, from the first emergency stage up until the complete aesthetic and functional rehabilitation”, and in the end of introduction section I wrote “The aim of this study is to define the best management of maxillofacial ballistic injuries and to describe a standardized, surgical, and prosthetic, rehabilitation protocol, from the first emergency stage up until the complete aesthetic and functional rehabilitation.”. In my opinion, it is the same concept but in different words because in the introduction, not having a limit of words, I could argue the choice of this study and the general characteristics of these complex traumas.

Point 3: The formatting of the paper is not in line with the journal template, for example: Headings should not be included in the abstract (e.g. Background:, Methods:); keywords should not be numbered; references are not properly formatted 

Response 3: thank you so much for your suggestion, I modified abstract and keyword as required. I also double-checked all the references.

Point 4: Add City/Country for Implant Manufacturers

Response 4: I’ve added it in Dental Rehabilitation Details, 3.3 section. 

Point 5: This article is too generally written; it is not obvious what are the elements from the case presentations that led to the conclusion and results. In addition, the treatment protocol is overly simplified.

Response 5: management of maxillofacial ballistic trauma, as widely underlined in this study, is complex topic, poor-represented in literature. The aim of our study is to present our experience regarding the treatment of these patients by defining a general management protocol, without going too deeply into the individual prosthetic phases. For this reason, we have decided to carry out a retrospective study. However, we do not exclude in the future the possibility of presenting a case report explaining more accurately the individual prosthetic phases of the protocol. In the results section, we used clinical cases to present our method of surgical and prosthetic patient’s management and to demonstrate that standardized and systematic protocol allows to treat this kind of patients in the same way, guaranteeing an effective result despite their complexity and diversity.

Point 6: Avoid repetitions throughout the text.

Response 6: I modified the article to reduce any ripetitions.

Point 7: Conclusions must be shortened, simplified and also must avoid repetitions.

Response 7: I modified the conclusion.

Round 2

Reviewer 1 Report

Dear Authors,

the authors have greatly improved the manuscript, but I can still highlight minor changes :

- lines 127 to 132 are criteria from the methodology and should be associated with table 1

- there is some confusion between the last paragraph of the discussion and the conclusions section. I believe that the authors want to highlight their experience in the follow-up, so I suggest removing "in conclusion" (line 380)

Author Response

Thank you for your suggestions!!

Point 1: lines 127 to 132 are criteria from the methodology and should be associated with table 1. 

Answer 1: We have already associated Brauner's classification with Table 1 in the section "Anatomic Sites Involved", where we have now specified the reference. 

Point 2: there is some confusion between the last paragraph of the discussion and the conclusions section. I believe that the authors want to highlight their experience in the follow-up, so I suggest removing "in conclusion" (line 380)

Answer 2: Yes, I modified it. 

Reviewer 3 Report

There are still some grammar mistakes in the text (e.g. "Maxillofacial ballistic injuries requires a multidisciplinary 400 and multistep approach"). Please check the text again. 

Please recheck the sections of the article to make sure the required structure is followed.  Currently, the last paragraph of discussions starts with "In conclusion".

The three cases are presented in extensive detail, but it is not clear how they are connected with the other cases for which no information is provided and what common patterns led to this protocol.

Author Response

Point 1: There are still some grammar mistakes in the text (e.g. "Maxillofacial ballistic injuries requires a multidisciplinary 400 and multistep approach"). Please check the text again. 

Answer 1: Thank you. We have rechecked again the text. 

Point 2: Please recheck the sections of the article to make sure the required structure is followed.  Currently, the last paragraph of discussions starts with "In conclusion".

Answer 2: Yes, I modified it. 

Point 3: The three cases are presented in extensive detail, but it is not clear how they are connected with the other cases for which no information is provided and what common patterns led to this protocol.

Answer 3: I modified the section "Dental rehabilitation" by reducing the discussions about single cases and clarifying the common prosthetic protocol. The three explanatory cases have been introduced to demonstrate the same prosthetic rehabilitation procedure, applied to all patients,  despite the different localization and type of residual defects after maxillofacial surgery.